# Role of *Leishmania infantum* in Meningoencephalitis of Unknown Origin in Dogs from a Canine Leishmaniosis Endemic Area

**DOI:** 10.3390/microorganisms9030571

**Published:** 2021-03-10

**Authors:** Miriam Portero, Guadalupe Miró, Rocío Checa, Elena Martínez de Merlo, Cristina Fragío, Miguel Benito, Ángel Sainz, Carmen Pérez

**Affiliations:** 1Department of Small Animal Medicine and Surgery, Hospital Clinico Veterinario Complutense, Veterinary Faculty, Universidad Complutense de Madrid, 28040 Madrid, Spain; emerlo@ucm.es (E.M.d.M.); cfa@ucm.es (C.F.); miguelbenitoben@gmail.com (M.B.); angelehr@ucm.es (Á.S.); cperezdiaz@vet.ucm.es (C.P.); 2Department of Animal Health, Veterinary Faculty, Universidad Complutense de Madrid, 28040 Madrid, Spain; rocichec@ucm.es

**Keywords:** cerebrospinal fluid, diagnosis, dog, IFAT, leishmania, meningoencephalitis of unknown origin, neurology, PCR

## Abstract

The main hypothesis for the aetiology of meningoencephalitis of unknown origin (MUO) in dogs is an autoimmune or genetic cause that is associated with a triggering event (environmental factors/infectious agents). The aim of this ambispective cohort study was to test for *Leishmania infantum* infection in the blood and cerebrospinal fluid (CSF) of dogs with MUO in an endemic area of canine leishmaniosis. Dogs with MUO were selected amongst all dogs undergoing blood anti-*L. infantum* antibody testing (control group). The blood plasma or serum samples from all dogs were analysed for anti- *L. infantum* antibodies by a quantitative indirect fluorescent assay (IFAT). In dogs with MUO, CSF samples were obtained for analysed by PCR detection of *L. infantum* DNA. Forty-four percent and 22% of the dogs in the MUO group featured magnetic resonance imaging (MRI) findings and CSF cytology respectively, consistent with *L. infantum* infection. IFAT, PCR, and histological findings were negative for *L. infantum.* A significant difference in *L. infantum* infection prevalence was found between the control and MUO group (*p* = 0.0022). While it seems unlikely that *L. infantum* plays a role in the aetiology of MUO, in endemic areas, this pathogen should be included in the differential diagnosis of this neurological disorder.

## 1. Introduction

Central nervous system (CNS) inflammatory diseases are perhaps the most frequent neurological disorders in dogs. Two broad groups have been defined: inflammatory diseases of infectious origin, in which an infectious agent is isolated as the disease cause, and those of a non-infectious aetiology [1].

Within the group of inflammatory non-infectious CNS diseases, the past few years have witnessed cases of necrotizing leukoencephalitis, granulomatous meningoencephalitis, and necrotizing meningoencephalitis, all with a diagnosis that is based on histopathological findings [2]. However, it is uncommon to obtain histopathological confirmation of these three conditions ante-mortem, and this has determined the creation of the term meningoencephalitis of unknown origin (MUO) to group together clinical signs, and magnetic resonance imaging (MRI), and/or computed tomography and/or cerebrospinal fluid (CSF) analysis findings that are compatible with a non-infectious inflammatory CNS disease [3].

Several studies have addressed potential prognostic factors for MUO [4,5,6] and its aetiology. The main hypothesis is an autoimmune or genetic cause [7,8] that is associated with a triggering factor that leads to the presentation of disease signs in a specific manner and at a given time point [9,10]. Among the different triggering factors examined so far are environmental factors and infectious agents as possible activating agents of CNS cells. Notwithstanding, no infectious agent has been isolated in this context [1,11,12,13,14,15,16].

Canine leishmaniosis is a systemic disease that is caused by the protozoan *Leishmania infantum*, which is endemic in many countries of the Mediterranean basin, such as Spain, where a high prevalence of this infection has been detected in the Madrid region [17,18]. Some authors argue that *L. infantum* is able to cross the blood-brain barrier. Once in the CSF, it then spreads, which causes an active CNS infection that is characterized by a non-suppurative granulomatous encephalomyelitis although peripheral nerves may also be affected [19,20]. The described neurological signs are varied and include seizures, blindness, vestibular and cerebellar signs, paraparesis, tetraparesis, tetraplegia, and myoclonus [21,22]. These clinical signs resemble those that were observed during the course of MUO [3,7,23].

The aim of the present study was to test for the presence of *L. infantum* infection in the CSF and blood of dogs with a diagnosis of MUO. Our main working hypothesis was that, in dogs in our geographical setting, which are highly exposed to this parasite of easy vector transmission, *L. infantum* infection could potentiate the immune response elicited by theses neurological diseases and, thus, aggravate the clinical picture in the absence of other pathogens (*Toxoplasma gondii* and *Neospora caninum*).

## 2. Materials and Methods

### 2.1. Animals

A clinical ambispective cohort study was performed over the period March 2010 to December 2020. The dogs were enrolled amongst those that were subjected to blood anti-*L. infantum* antibody determination using a quantitative serological indirect immunofluorescence antibody test (IFAT) from March 2010 to April 2017 (control group). Within this control group, the dogs under a diagnostic protocol for MUO were selected according to the following inclusion criteria (MUO group):

(1) A diagnosis of MUO that is based on clinical presentation (focal or multifocal signs of a CNS abnormality detected in the neurological exam), alterations detected in an MRI study (hyperintense lesions in T2-weighted images), and/or CSF analysis (hyperproteinorrachia and/or lymphocytic, monocytic or mixed pleocytosis).

(2) Medical records indicating the presence of canine distemper virus, *Toxoplasma gondii*, and *Neospora caninum* in CSF determined by PCR.

(3) *L. infantum* DNA in CSF (nested PCR detection) determined in dogs when a frozen CSF sample was available.

(4) Patient data available on the clinical course of disease at the study end.

Owner informed consent was obtained from all clinical cases, and all procedures performed adhered to ethical, welfare, and data protection standards. Dogs with a poor clinical status were euthanized through an intravenous injection of pentobarbital sodium.

### 2.2. Imaging

MRI (Panorama 0.23T (Philips Medical Systems Nederland B.V, Eindhoven, The Netherlands) and CSF collection was performed with the dog under general anaesthesia. The pulse sequences included sagittal and transverse T2-weighted images, transverse T2-fluid-attenuated inversion recovery images, transverse Fast Field Echo tridimensional T2-weight images, and transverse T1-weighted images before and after paramagnetic contrast injection of gadolinium (0.1 mmol/kg intravenous) (Prohance Sol. 279.3 mg/mL, Laboratorios Rovi, Madrid, Spain)

### 2.3. CSF Collection

Following MRI, CSF was collected by the puncture of the cerebellomedular cistern using an aseptic technique into tubes containing EDTA. Aliquots for cytological study were processed immediately while samples for the detection of infectious diseases were frozen at −40 °C for subsequent analysis. The CSF samples contaminated with blood (>5000 red blood cells/μL) were discarded [24,25]. The results were recorded as abnormal when protein concentrations were >0.3 g/L and/or nucleated cell counts were >5 cells/µL [26].

### 2.4. Immunologic Blood Tests

Blood plasma or serum samples were tested for antibodies by IFAT. The antigen was obtained from a culture of promastigotes of *L. infantum* L-75 set up in Novi, McNeal, and Nicolle medium. The plates were read under an epifluorescence microscope Olympus BH-2 (Olympus Imaging America Inc., Center Valley, PA, USA) with a blue filter using the x400 objective. The antibody titre cut-off established for a positive result was ≥1/100 [27].

### 2.5. Molecular Diagnosis

The presence of *L. infantum* DNA in CSF was examined by PCR. The frozen CSF samples were thawed at room temperature (total volumes ranging from 0.5–1 mL). The QIAamp DNA Mini Kit (QIAGEN, Venlo, The Netherlands) was used to extract DNA according to the manufacturer’s instructions with the following modifications [28]:

In order to obtain the initial volume, 100 µL were aspirated from the supernatant after centrifugation of the original sample (0.5–1 mL of CSF); 20 µL of proteinase K, 100 µL of ATL buffer, and 200 µL of AL buffer were added, and the mixture was incubated for 10 min (min.) at 56 °C. Thereafter, the procedure was continued, as described by the manufacturer. The elution volume was 200 µL.

The purified DNA was stored at −20 °C until its later use.

The polymerase chain reaction was performed via *Leishmania* genus specific nested-PCR of the variable region of the gene coding for SSUrRNA (adapted from Cruz et al., (2002) [28]) using the primers: R1(R221): 5′-GGTTCCTTTCCTGATTTACG-3′; R2 (R332): 5′-GGCCGGTAAAGGCCGAATAG-3′; R3 (R223): 5′-TCCCATCGCAACCTCGGTT-3′; and R4 (R333) 5′-AAAGCGGGCGCGGTGCTG-3.

For the first amplification: 20 µL of DNA was added to 30 µL of master mix containing 15 pmol of the primers R221 and R332 [29]; 200 µM dNTP mix, 5 µL de buffer 10X, 2 mM MgCl_2_, and 1.4 units of Tth DNA polymerase (1 U/µL) (Biotools, Madrid, Spain). The thermal cycler (2720, Applied Biosystems, Madrid, Spain) conditions were 80 °C for 2 min., followed by 94 °C for 5 min. This was followed by 30 cycles of: 94 °C for 30 s (s), 60 °C for 30 s, an extension stage at 72 °C for 30 s, followed by a final extension at 72 °C for 10 min. The samples of DNA detected in the size band of 603 base pairs (bp) were considered to be positive for *Leishmania*.

For the second amplification: 10 µL of a l/40 dilution in sterilized distilled water of the product of the first amplification was added to 15 µL of the master mix volume containing 7.5 pmol and 3 pmol of each specific primer R223 and R333, respectively [29]; 200 µM dNTP mix, 2.5 µL de buffer 10X, 2 mM MgCl_2_, and 0.7 µL of Tth DNA polymerase (1 U/µL) (Biotools, Madrid, Spain). The thermal cycler program was 80 °C for 2 min., 94 °C for 5 min., and then 30 cycles at 94 °C for 30 s, 65 °C for 30 s, and 72 °C for 30 s; this was followed an extension step at 72 °C for 5 min. Samples of size 353 bp were taken as positive for the DNA of *Leishmania*.

All of the amplified products were resolved on a 1.5% agarose gel containing SYBR Safe Gel Stain (Invitrogen, CA, USA) and then visualized with a dark reader transilluminator (Clare Chemicals, Dolores, CO, USA).

### 2.6. Data Collected from Medical Records

Data regarding clinical signs, neurological exam, haemogram and biochemical profiles prior to diagnosis (when available), MRI and CSF findings, treatment, and outcome were recorded.

### 2.7. Statistical Analysis

All of the analyses were performed using the Statistical Package for Social Science (SPSS) version 22 (SPSS, Chicago, IL, USA) and SAS version 9.4 (SAS Institute, Cary, NC, USA). A descriptive analysis of the MUO group is provided, including means, maxima, and minima of the quantitative variables, as well as the proportions of dogs with different neurological clinical signs, along with MRI, CSF, blood IFAT, and CSF PCR findings. The prevalence of *L. infantum* infection in the control and MUO groups was compared using Fisher´s test. Significance was set at *p* < 0.05.

## 3. Results

### 3.1. Study Population

During the study period, 2312 dogs were subjected to the detection of anti-*L. infantum* antibody by IFAT. Of these dogs, 68 were entered into a diagnostic protocol for MUO, such that the control group consisted of 2244 dogs and the MUO group of 68 dogs. The MUO group comprised 30 female and 38 male dogs of a mean age 6.2 years (range six months to 13 years) and mean weight 8.5 kg (range 1.2 to 47.7 kg). The breeds included were Yorkshire terrier (*n* = 19), Mongrel (*n* = 13), French bulldog (*n* = 10), Bichon Maltese (*n* = 7), Bichon Frise (*n* = 3), Doberman Pinscher (*n* = 3), Pomeranian (*n* = 2), Pug (*n* = 2), Poodle (*n* = 2), Shih-tzu (*n* = 1), Chihuahua (*n* = 1), German Shepherd (*n* = 1), Jack Russel (*n* = 1), Labrador retriever (*n* = 1), Miniature Schnauzer (*n* = 1), and Rottweiler (*n* = 1).

### 3.2. Clinical Signs

Table 1 provides the main neurological signs observed.

### 3.3. Laboratory Tests

Table 2 presents the lab test results.

### 3.4. Imaging Findings (N = 68/68)

MRI was performed in all dogs that were assigned to the MUO group. MRI lesions were observed in 88% of the dogs: multifocal (47%), focal (35%), or diffuse (18%). The remaining 12% had no alterations on MRI. The affected regions were: brain (76.7%), brainstem (33.3%), spinal cord granulomas (31.7%), and cerebellum (10%). Other MRI findings were: perilesional contrast (45%), ventriculomegalia (38.3%), mass effect (35%), syringomielia (33.3%), perilesional oedema (30%), loss of brain sulci (25%), midline displacement (23.3%), meningeal contrast enhancement (16%), and cerebellar herniation (11.7%). No affected peripheral nerves were found in this study.

### 3.5. CSF Analysis (N = 51/68)

Cerebrospinal fluid collection was contraindicated in 17 dogs due to signs of increased intracranial pressure. The mean CSF protein was 0.50 g/L (range 0.17 to 1.22 g/L). Mean total nucleated cell count was 107 cells µL/(range 0 to 1720 cells/µL). The most frequently observed pleocytosis was lymphocytic (33%), monocytic (22%), followed by mixed (16%). In 29% of CSF samples, no pleocytosis was observed.

Canine distemper virus, *Toxoplasma gondii*, and *Neospora caninum* in CSF were determined by PCR in 45 of 51 dogs undergoing CSF analysis. All of the dogs returned a negative result.

### 3.6. Immunologic Blood Tests

Control group: 235/2244 Dogs tested positive for *L. infantum* according to the blood plasma or serum IFAT.

MUO group: No Dogs (0/68) in the MUO Group Tested Positive for *L. infantum* According to the Blood Plasma or Serum IFAT.

### 3.7. CSF Moleculardiagnosis in the MUO Group (N = 45/51)

Frozen CSF samples were available for 45 of 51 dogs that were subjected to CSF analysis. All of these dogs tested PCR negative for *L. infantum* (Figure 1 and Figure 2).

### 3.8. Treatment and Outcome

Thirty-four of the 68 dogs were treated with prednisolone in combination with subcutaneous cytosine arabinoside (Citarabina; Pfizer, Madrid, Spain), 22 were only treated with prednisolone (22/45) and six were treated with prednisolone in combination with cyclosporine (Atopica; Novartis, Barcelona, Spain) (4/45). Six dogs (6/45) did not receive specific MUO treatment, because they had been euthanized or had died at the time of the diagnosis or in the following 24 h due to their poor clinical status.

The median survival time was 1030 days (range 0 to 3520 days). Twenty-two dogs (22/68) were alive at the end of the study. Of the 46 dogs that died, 13 died from causes that were unattributable to MUO. A complete ordered systematic necropsy was only possible in 10 dogs. In three dogs, granulomatous meningoencephalitis, and in seven dogs, leukonecrotizing encephalitis, were histologically confirmed. In the 10 necropsies, the presence of amastigotes of *L. infantum* was not detected in the tissues analysed, including CNS tissue.

### 3.9. Statistical Analysis

The prevalence of *L. infantum* was 10.47% in the control group and 0% in the MUO group, with the difference being significant *p* = 0.0022.

## 4. Discussion

Many hypotheses have been proposed in order to explain the aetiology of MUO, including autoimmune disease, genetic predisposition, direct CNS infection, and parainfectious immune dysregulation. However, the theory with most support to date is a multifactorial origin, involving both genetic and environmental factors [7,11,15,30]. The potential involvement of an infectious agent playing a decisive role in MUO has prompted the implementation of routine tests that were designed to detect these pathogens in affected dogs. In addition, as the only effective treatment described in dogs with MUO is immunosuppressive therapy, this determines a need to assess possible underlying infectious diseases, whose progression could be jeopardized by this treatment of the neurological disease [3]. As far as we know, this is the first study to examine the presence of *L. infantum* infection in the CSF of dogs that were diagnosed with MUO living in areas that are endemic for canine leishmaniosis.

Giannuzzi et al. (2017) [20] examined ten dogs that were naturally infected with *L. infantum*. Four out of the ten dogs showed clinical neurological signs [20]. However, it is known that clinically healthy dogs with natural *L. infantum* infection can show CNS alterations as the consequence of mononuclear infiltration. *L. infantum* triggers numerous mechanisms in the host, many of which remain unknown, which enable the entry of this parasite into the CNS with no preference being shown for any specific region [21]. Among the many presentations, we should highlight lymphoplasmacytic meningoencephalitis with involvement of the choroid plexus and/or leptomeninges [19,31,32,33,34,35], granulomas in the spinal cord [19,20,36,37,38] and affected peripheral nerves [20,38]. In the present study, 43% of dogs with MUO showed meningeal involvement and similar medullary lesions as those that were described in dogs with natural *L. infantum* infection. In zones where canine leishmaniosis is endemic, we feel a differential diagnosis of this protozoosis is needed based on specific tests of high sensitivity that will help to distinguish whether any CNS lesions found could be the consequence of infection or exclusively attributable to MUO.

Other neurological alterations described in canine leishmaniosis are cerebral infarcts in blood vessel walls secondary to vasculitis due to the deposition of immunocomplexes [20,39,40] and convulsions due to blood hyperviscosity [41]. None of our dogs had cerebral infarct or blood hyperviscosity.

Magnetic resonance imaging or CSF analysis in all of the dogs examined in this study were poorly specific of infection. The most frequent CSF cell pattern detected was lymphocytic pleocytosis (33%), which is consistent with reported findings in dogs with MUO [3,42,43]. Notwithstanding, 22% of dogs show similarly elevated monocyte counts to those that were observed in individuals with leishmaniosis and neurological abnormalities [20,22].

The search for *L. infantum* amastigotes in the CNS of infected dogs is fairly infrequent [33,34,36,38]. However, in most dogs with nonsuppurative meningoencephalomielitis secondary to the infection, the PCR detection of *L. infantum* DNA in the CNS or CSF usually proves positive, which suggests the high diagnostic sensitivity of this technique [19,38,44,45,46]. In our study, all sick dogs tested IFAT seronegative. However, we cannot completely rule out a lack of CNS infection, as this could have occurred in the absence of an active humoral response. Thus, it is essential to exclude a role of *L. infantum*, mainly through a molecular diagnosis that is based on the specific detection of DNA in samples of CSF and/or cerebral or medullary tissue.

The significant difference in MUO prevalence found in our control and MUO groups and our lack of identification of infection through very sensitive techniques in a population highly exposed to *L. infantum* indicates that it is unlikely that this agent participates in the aetiology of MUO or acts as a trigger event, as described for other agents, such as distemper virus, *Toxoplasma gondii*, and/or *Neospora caninum* [1,11,12,13,14,15,16]. Although further studies with a larger number of dogs are needed, it should be highlighted that the infectious agent looked for was ruled out in every dog included in this study. Despite this, we believe it is important to continue the search for a possible role of *L. infantum* infection in dogs with MUO, especially in zones where canine leishmaniosis is endemic.

Future studies need to confirm MUO through histopathology and immunohistochemistry to try identify the presence of amastigotes of *L. infantum* in haematopoietic organs and nerve tissue, although, at present, this is considered to be infrequent.

## 5. Conclusions

Although 44% and 22% of the dogs examined in this study featured MRI and CSF cytology findings, respectively, consistent with *L. infantum* infection, quantitative IFAT serology, PCR, and histology findings were all negative for *L. infantum.*

It seems unlikely that *L. infantum* plays a role in the aetiology of MUO, as significant differences were found in the prevalence of *L. infantum* between our control and MUO groups.

When considering that a large proportion of the dogs showed similar findings to those described in dogs with clinical leishmaniosis, we strongly recommend that this parasitic zoonotic disease is included in the differential diagnosis of any neurological syndrome in dogs inhabiting endemic zones of canine leishmaniosis, especially when the treatment of choice is an immunosuppressive agent.

## Figures and Tables

**Figure 1 microorganisms-09-00571-f001:**
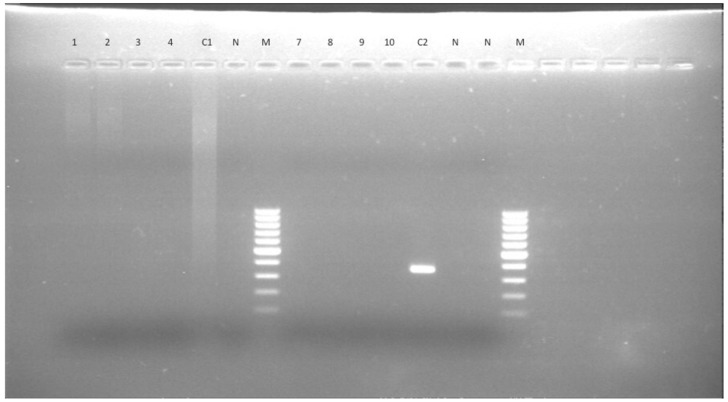
*Leishmania* genus specific nested-PCR for the diagnosis of *L. infantum* infection in dogs. cerebrospinal fluid (CSF) negative results: lane 1 to 4, first amplification; lane 7 to 10, second nested amplification. Lane C1 and C2 are positive controls (*L. infantum* DNA) for the first and second amplification, respectively. Lane N, negative controls (no DNA). M = molecular size marker (100–1000 bp).

**Figure 2 microorganisms-09-00571-f002:**
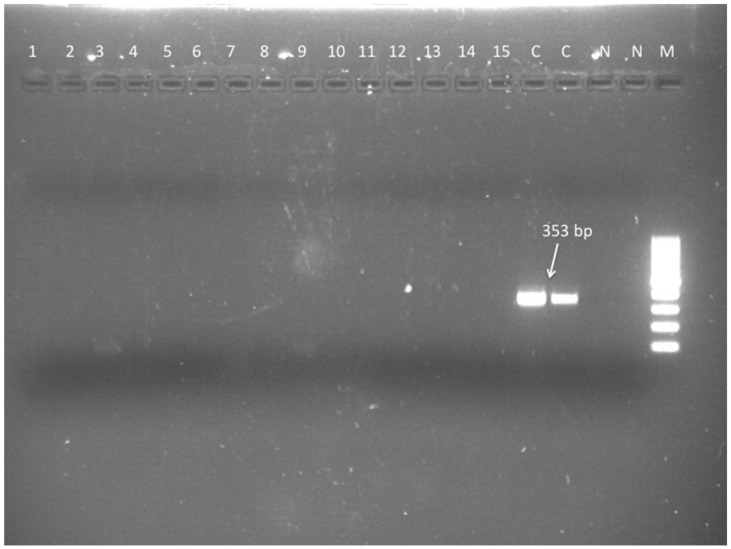
*Leishmania* genus specific nested-PCR for the diagnosis of *L. infantum* infection in dogs. Second nested amplification. M = molecular size marker (100–1000 bp). Lane 1 to 15 negative results (CSF DNA). Lanes C positive controls (*L. infantum* DNA). Lanes N, negative controls (no DNA).

**Table 1 microorganisms-09-00571-t001:** Neurological signs observed in dogs in the meningoencephalitis of unknown origin (MUO) group.

Clinical Signs	%
Vestibular disorders	50.8
Seizures	34.9
Impaired vision	31.8
Circling	31.8
Altered mental state	26.9
Cervical pain	26.9
Cerebellar alterations	23.8
Aggressiveness	17.5
Paraparesis	15.8
Hemiparesis	14.28
Tetraparesis	12.7
Head pressing	6.3

**Table 2 microorganisms-09-00571-t002:** Laboratory findings in dogs in the meningoencephalitis of unknown origin (MUO) group (*N* = 68).

Parameter	*N* ^1^	Mean	Range
Haematocrit (L/L)	65	0.46	0.36–0.61
Total solids (g/L)	65	66.6	49–82
White blood cells (×10^9^/L)	65	10.6	4.8–26.4
Neutrophils (×10^9^/L)	65	7.7	3.4–24.2
Band neutrophils (×10^9^/L)	65	0.74	0–1.1
Glucose (mmol/L)	63	4.16	3.3–6.11
Creatinine (µmol/L)	63	106.10	61.8–123.7
Urea (mmol/L)	63	7.49	3.3–9.9
ALT ^2^ (µkat/L)	63	0.83	0.33–1

^1^*N*: sample size; ^2^ ALT: alanine aminotransferase.

## Data Availability

The data presented in this study are available on request from the corresponding author.

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
