# Peer review of "Role of Leishmania infantum in Meningoencephalitis of Unknown Origin in Dogs from a Canine Leishmaniosis Endemic Area"

_microorganisms, 2021, doi:10.3390/microorganisms9030571_

Round 1
Reviewer 1 Report
The study examines an interesting hypothesis i.e. the potential association of canine leishmaniasis with meningoencephalitis, or L. infantum as an etiological factor of meningoencephalitis. Although this hypothesis is very interesting, there are several confusing parts in the manuscript that have to be extensively revised, the majority of which are mentioned below. Also, there are major parts of the manuscript that need better English editing.
line 17: do you mean leishmaniasis endemic area? Please clarify
While up to the end of the introduction the authors present data concerning the MUO in dogs, in the proposed hypothesis (lines 61-64) they suddenly mention patients. Do they mean human patients? If so, they should add a more extensive explanation of how they came up to this hypothesis. If no, they should replace by another term.
This is also a problem in the whole manuscript. The reader is often confused between the dogs and the patients. I suggest to replace and clarify the terms everywhere in the manuscript
Similarly, again in Materials and Methods, the selection of the examination of criteria regarding Toxoplasma gondii and Neospora caninum presence in CSF have not been justified previously. The authors should explain in the Introduction why they included this analyses, in combination with the main hypothesis.
line 107: Since, as reported in the results, no L. infantum was detected in any tested sample, the term “was detected” should be replaced by “L. infantum presence was examined”
line 109: Not QUIAGEN, QIAGEN
line 109: replace “were” by “was”
I believe lines 153-155 are from the template and should be deleted
lines 157-165: The majority of this part belongs to Materials and Methods and should be moved there
Since Figures 1 and 2 are referred in line 199, it is not convenient to constitute a separate section (3.10, line 217)
lines 252-253: This result is firstly mentioned here in the Discussion. It should be also in the Results section
Author Response
The authors thank the reviewers their comments which will improve the manuscript.
Note: the line numbers are no longer the same, since the order of the whole text has been modified
This new version of the text has been checked over again by Ana Burton who was responsible for translating the original article from Spanish.
Point 1: line 17: do you mean leishmaniasis endemic area? Please clarify
Response 1: Yes, the study has been performed in a leishmaniosis endemic area. The title (lines 3-4) and line 31 have been modified to clarify this point.
Point 2: While up to the end of the introduction the authors present data concerning the MUO in dogs, in the proposed hypothesis (lines 61-64) they suddenly mention patients. Do they mean human patients? If so, they should add a more extensive explanation of how they came up to this hypothesis. If no, they should replace by another term.
This is also a problem in the whole manuscript. The reader is often confused between the dogs and the patients. I suggest to replace and clarify the terms everywhere in the manuscript
Response 2: all the “patients” included in this study were dogs. In order to avoid confusion the term “patients” has been replaced by dogs in the whole manuscript.
Point 3: Similarly, again in Materials and Methods, the selection of the examination of criteria regarding Toxoplasma gondii and Neospora caninum presence in CSF have not been justified previously. The authors should explain in the Introduction why they included this analyses, in combination with the main hypothesis.
Response 3: The lines 80-81 have been modified to explain why Toxoplasma gondii and Neospora caninum were included in the analyses. “Our main working hypothesis was that, in dogs in our geographical setting, which are highly exposed to this parasite of easy vector transmission, L. infantum infection could potentiate the immune response elicited by theses neurological diseases and thus aggravate the clinical picture in absence of other pathogens (Toxoplasma gondii and Neospora caninum).”
Point 4: line 107: Since, as reported in the results, no L. infantum was detected in any tested sample, the term “was detected” should be replaced by “L. infantum presence was examined”
Response 4: revised as requested. The line 125 has been modified: “The presence of L. infantum DNA in CSF was examined by PCR”
Point 5: line 109: Not QUIAGEN, QIAGEN
Response 5: revised as requested. Line 127
Point 6: line 109: replace “were” by “was”
Response 6: revised as requested. Line 127
Point 7: I believe lines 153-155 are from the template and should be deleted
Response 7: Yes, it is a transcription error. Lines 171-173 have been deleted.
Point 8: lines 157-165: The majority of this part belongs to Materials and Methods and should be moved there
Response 8: the lines 175-184 describe the demographic results of this study, and for this reason these data were included in the “Results” section:
“During the study period 2312 dogs were subjected to detection of anti-L. infantum antibody by IFAT. Of these dogs, 68 were entered into a diagnostic protocol for MUO such that the control group consisted of 2244 dogs and the MUO group of 68 dogs. The MUO group comprised 30 female and 38 male dogs of mean age 6.2 years (range 6 months to 13 years) and mean weight 8.5 kg (range 1.2 to 47.7 kg). The breeds included were Yorkshire terrier (n=19), Mongrel (n=13), French bulldog (n=10), Bichon Maltese (n=7), Bichon Frise (n=3), Doberman Pinscher (n=3), Pomeranian (n=2), Pug (n=2), Poodle (n=2), Shih-tzu (n=1), Chihuahua (n=1), German Shepherd (n=1), Jack Russel (n=1), Labrador retriever (n=1), Miniature Schnauzer (n=1) and Rottweiler (n=1).”
Demographic data are the data regarding the signalment of the studied population such as age, breed, sex and origin. For that reason, we included this part in “Results” section instead of Materials and Methods. However, if the reviewer considers that the manuscript will improve by moving this paragraph to Materials and Methods the authors will do it
Point 9: Since Figures 1 and 2 are referred in line 199, it is not convenient to constitute a separate section (3.10, line 217)
Response 9: revised as requested. In the line 254 the section 3.10 has been deleted
Point 10: lines 252-253: This result is firstly mentioned here in the Discussion. It should be also in the Results section
Response 10: The authors have doubts about Point 10. The lines 288-290 are: “presence of granulomas in the spinal cord (Cauduro et al., 2011; Márquez et al., 2013; José-López et al., 2014; Da Costa Oliveira et al., 2017; Giannuzzi et al., 2017), and affected peripheral nerves [20,38]”. This sentence is not a result of our study, it is part of the discussion referring to other studies.
In lines 209-210 we describe: “Affected regions were: brain (76.7%), brainstem (33.3%), spinal cord (31.7%) and cerebellum (10%).” The presence of granulomas in the spinal cord as mentioned in lines 288-290 corresponds to the 31.7% spinal cord affection in our study. No affected peripheral nerves were found in our study.
The authors suggest the following modifications in lines 209-214 to clarify this point: “Affected regions were: brain (76.7%), brainstem (33.3%), spinal cord granulomas (31.7%) and cerebellum (10%). Other MRI findings were: perilesional contrast (45%), ventriculomegalia (38.3%), mass effect (35%), syringomielia (33.3%), perilesional oedema (30%), loss of brain sulci (25%), midline displacement (23.3%), meningeal contrast enhancement (16%) and cerebellar herniation (11.7%). No affected peripheral nerves were found in this study.”
Reviewer 2 Report
In this paper, the Authors evaluate the presence and the role of Leishmania infantum in meningoencephalitis of un-known origin in dogs from an endemic area
The Authors have investigated an interesting topic.
The manuscript is well written, presented and discussed.
In general, the structure of the article is satisfactory and in agreement with the journal instructions for authors.
The objectives of the paper are of interest and fit well within the scope of the journal.
Minor revisions are required:
Line 54: lacks reference.
Line 166: clinical signs could be presented in table.
Line 326: The references are not described as request in author instructions. For example: Tipold, A. TTLE. Abbreviated Journal Name in italics, Year in bold, after volume put a comma.
In my opinion, the manuscript could be accepted for publication in Microorganisms with the above-mentioned suggestions.
Author Response
The authors thank the reviewer her/his comments which will improve the manuscript.
Note: the line numbers are no longer the same, since the order of the whole text has been modified
This new version of the text has been checked over again by Ana Burton who was responsible for translating the original article from Spanish.
Point 1: Line 54: lacks reference.
Response 1: the references for the line 69 are the same for the second phrase in line 72 and have been mentioned at the end of the second phrase. “Some authors argue that L. infantum is able to cross the blood-brain barrier. Once in the CSF it then spreads causing an active CNS infection characterized by a non-suppurative granulomatous encephalomyelitis although peripheral nerves may also be affected [19,20]”
Point 2: Line 166: clinical signs could be presented in table.
Response 2: revised as requested. Lines 187-190 have been modified and a new table has been added in lines 192- 202 (Table 1. Neurological signs observed in dogs in the MUO group). Note that previous Table 1 now is Table 2.
Point 3: Line 326: The references are not described as request in author instructions. For example: Tipold, A. TTLE. Abbreviated Journal Name in italics, Year in bold, after volume put a comma.
Response 3: all references have been modified according to the author instructions. Lines 374- 478.
Round 2
Reviewer 1 Report
I am generally satisfied with the modifications conducted and I believe the manuscript has been improved and should be published